# Photoinduced Processes in Lysine-Tryptophan-Lysine Tripeptide with L and D Tryptophan

**DOI:** 10.3390/ijms24043331

**Published:** 2023-02-07

**Authors:** Aleksandra A. Ageeva, Roman S. Lukyanov, Sofia O. Martyanova, Ilya M. Magin, Alexander I. Kruppa, Nikolay E. Polyakov, Victor F. Plyusnin, Alexander B. Doktorov, Tatyana V. Leshina

**Affiliations:** 1V.V. Voevodsky Institute of Chemical Kinetics and Combustion, Siberian Branch of the Russian Academy of Sciences, 3 Institutskaya Str., 630090 Novosibirsk, Russia; 2Department of Natural Sciences, Department of Physics, Novosibirsk State University, 2 Pirogova Str., 630090 Novosibirsk, Russia

**Keywords:** peptides, optical isomers, radical-ions, biradical-zwitterions, fluorescence, NMR, chemical polarization, UV-irradiation

## Abstract

Optical isomers of short peptide Lysine-Tryptophan-Lysine (Lys-{L/D-Trp}-Lys) and Lys-Trp-Lys with an acetate counter-ion were used to study photoinduced intramolecular and intermolecular processes of interest in photobiology. A comparison of L- and D-amino acid reactivity is also the focus of scientists’ attention in various specialties because today, the presence of amyloid proteins with D-amino acids in the human brain is considered one of the leading causes of Alzheimer’s disease. Since aggregated amyloids, mainly Aβ42, are highly disordered peptides that cannot be studied with traditional NMR and X-ray techniques, it is trending to explore the reasons for differences between L- and D-amino acids using short peptides, as in our article. Using NMR, chemically induced dynamic nuclear polarization (CIDNP) and fluorescence techniques allowed us to detect the influence of tryptophan (Trp) optical configuration on the peptides fluorescence quantum yields, bimolecular quenching rates of Trp excited state, and the photocleavage products formation. Thus, compared with the D-analog, the L-isomer shows a greater Trp excited state quenching efficiency with the electron transfer (ET) mechanism. There are experimental confirmations of the hypothesis about photoinduced ET between Trp and the CONH peptide bond, as well as between Trp and another amide group.

## 1. Introduction

The effect of light exposure on proteins is attracting attention due to the widespread use of the latter in practice [1,2,3,4,5,6]. These are photobiology, medicine, and the research of protein reactions, including photoinduced polymerization. For short peptides, the main pathways of photoinduced transformations have already been established [7,8,9] (Figure 1).

The use of several physical methods, including ESR, has led to the assumption that most cleavage pathways are radical and, in some cases, begin with electron transfer (ET), resulting in the formation of a biradical-zwitterion (BZ). Thus, according to [8], cyclization of the glycyltryptophan occurs in BZ (Figure 2).

In addition, much attention is paid to the role of ET in photophysical processes in proteins, e.g., in the quenching of tryptophan (Trp) fluorescence. It is well known that Trp fluorescence is highly sensitive to the microenvironment and is widely used to study the structure of proteins [10,11,12]. Two quenching mechanisms are generally considered: Förster energy transfer and ET [2]. However, there are no direct observations of ET between amino acid fragments and the peptide bond CONH. In addition, Figure 1 and Figure 2 proposed in the literature remain hypothetical. Therefore, an indirect but highly sensitive method for detecting paramagnetic species in the reaction—chemically induced dynamic nuclear polarization (CIDNP) seems appropriate.

The CIDNP phenomenon manifests in the NMR spectra of products formed from a pair of radicals with a non-Boltzmann population of nuclear spin sublevels [13]. An enhanced signal absorption or emission is observed when the reaction is carried out directly in the probe of the NMR spectrometer. An analysis of polarized signals allows one to obtain “portraits” of the paramagnetic precursors of the products. Recently, CIDNP has demonstrated a new phenomenon—spin selectivity in the processes of photoinduced ET in chiral systems. Spin selectivity is a difference in the CIDNP enhancement coefficients for diastereomers with different optical configurations [14,15]. This finding allows us to expect that the CIDNP study of photoinduced processes in short peptides on the examples of Lysine-Tryptophan-Lysine (Lys-Trp-Lys) acetate salt and Lys-{L/D-Trp}-Lys with N-terminal acetylation will provide answers to some crucial questions. First, this is an experimental confirmation of the hypothesis of ET between amino acids in peptides and the CONH peptide bonds. Second, this is the exploration of differences in the reactivity of systems with L- and D-amino acids [12]. It is crucial since replacing L-amino acids with D-analogs during aging causes Alzheimer’s and Parkinson’s diseases, type 2 diabetes, and several other ailments [16,17,18]. The presence of D-isomers of amino acids has been shown to result in disturbances in folding processes, leading to the aggregation of highly disordered amyloid proteins and peptides, forming large oligomers or fibrils, which have a toxic effect and destroy the brain. The current trend in studying the difference between L- and D-amino acids in highly disordered proteins and peptides is using short peptides as an example [17,18]. This approach to solving this problem is proposed in the literature since highly disordered proteins cannot be studied using high-resolution magnetic resonance and X-ray spectroscopy [17]. In this regard, an investigation of short peptides containing L- or D-Trp residues allows us to trace the differences between the reactivity of its optical isomers since Lys-{L/D-Trp}-Lys serves as a convenient model due to the strong fluorescence of Trp residue. Thus, this article will investigate the physical and chemical pathways of degradation of the Trp excited states in the composition of the peptides using ^1^H NMR, CIDNP, and fluorescence techniques.

## 2. Results and Discussion

Next, we will consider sequentially the photophysical and photochemical processes occurring during the quenching of the excited states of Trp in peptides **I**–**III** (Figure 1).

### 2.1. Fluorescence Quenching Processes in Lys-Trp-Lys

The emission spectra of **I**–**III** compared with L-N-acetyltryptophan (L-NAcTrp) in an aqueous solution with corresponding quantum yields and fluorescence decay traces are shown in Figure 2 and Table 1.

The fluorescence decay traces of the peptides **I**–**III** are described with two exponential models (see Table 1). As a rule, the multiexponential fluorescence decay of Trp, as part of a protein and peptide, was associated with short-lived and long-lived rotamers of Trp. There are a several models proposed to explain multi-exponential fluorescence decay [11].

As can be seen from Table 1, the highest fluorescence quantum yield (i.e., the lowest efficiency of quenching of the Trp singlet excited state) is observed for the D-optical isomer of peptide **II**. Moreover, there is a remarkable difference in the distribution of fluorescence decay times A_1_ and A_2_ in **I**–**III**. This seems to be the result of different contributions of short-lived rotamers in the isomers of peptides.

### 2.2. ^1^H NMR Investigation of Lys-Trp-Lys Photolysis Products

The ^1^H NMR spectra of peptides’ aqueous solutions are shown in Figure 3 and Appendix A. UV irradiation of aqueous solutions of peptides was carried out directly in the NMR spectrometer probe, making it possible to follow the formation of products and detect radical species using the CIDNP technique [13,19].

After durational UV irradiation, product signals are observed only for peptide **III** (Figure 3). A quantitative analysis of the intensities ratios in the NMR spectra shows that products in cases **I** and **II** are also formed. Nevertheless, their signals overlap with the signals of the initial peptides (see Appendix A). To detect product signals, we compared the integral signal intensities of **I**–**III** in solutions before and after photolysis and the ratios of individual lines in the reaction mixture with those in the starting compounds before photolysis. Appendix A with the analysis of deviations in the balance of signal intensities from those for the initial **I**–**III** are given in Appendix A. These deviations reflect the process of peptide cleavage. Similar changes in the intensity of the NMR signals of different protons indicate the formation of a single product and aid in establishing its possible structure. An analysis of these changes in the integral intensities shows that the degree of photocleavage is minimal for peptide **I**; on average, it does not exceed 15% relative to the signal intensity of the initial peptide with an accuracy of 10%. The maximum change is shown by proton 1, located in the central part of the molecule. This change points to the decarboxylation process, which slightly alters the position of the lines in the product compared to the initial one (Figure 3).

The most significant changes in the intensities of NMR signals are characteristic for peptides **II** and **III**. Changes in the integral intensity of peptide **II** mainly concern the protons of the Trp moiety (7, 8–12), 1, 1′ and the N-acetyl group, and as in the case of **I**, do not increase with increasing concentration from 1 to 8 mM (Appendix A). The latter indicates that the products were mainly formed in an intramolecular process. On the other hand, according to the general scheme of photoinduced radical processes typical for peptides given in the Introduction, a change in the protons intensities of the Trp moiety and 1, 1′ indicates the detachment of the Trp sidechain and cleavage of peptide bonds. It can be assumed that photoinduced radical cleavage of the C^6^–C^7^ bond in peptide **II** results in yielding the main product—3-methyleneindolenine. Simulation of the NMR spectrum of 3-methyleneindolenine does not contradict this assumption (Appendix A). The second product could be a dimer formed from peptide residues (Figure 4). The participation of radicals in forming dimers of peptides and other proteins under UV irradiation is widely described in the literature [4,5,6].

At the same time, the cleavage of the peptide bond is likely to result in the formation of an aldehyde in the case of peptides **II** and **III** (see Appendix A). The scheme describing the possible way of aldehyde formation is also presented in Appendix A.

As for the acetate salt of peptide **III**, the maximum changes in the NMR signal intensities concern 12, 9 aromatic protons and 1′, 1, 7 aliphatic protons (Appendix A). In addition, during intensive photolysis, product signals accumulate in the region of aromatic protons resonance (Figure 3). Simultaneous changes in the 12, 9, and 1 proton signals suggest forming a cycle similar to that previously described for the glycine-L-tryptophan peptide in [8] (Figure 2 and Figure 5). According to the authors of [8], under UV irradiation, the glycine-L-tryptophan peptide cyclization product is formed through the biradical-zwitterion (BZ) result of the intramolecular ET. The results of the NMR spectrum simulation (Appendix A) confirm the possibility of forming such a cycle from peptide **III**.

Moreover, a marked change in the intensities of protons 1′ of peptide **III** indicates the possible cleavage of the peptide bond, leading to the formation of an aldehyde (Appendix A).

### 2.3. Photo-CIDNP Study of Lys-Trp-Lys

In this part, the CIDNP study was used to detect the radical stages postulated in Figure 3, Figure 4 and Figure 5. However, this study has shown that in time-resolved (TR) photo-CIDNP spectra demonstrating intramolecular processes, the effects are absent for all peptides, whereas CIDNP effects appear when using the pseudo-steady state (PSS) experimental technique [19] (Figure 4).

In this technique, the CIDNP spectra reflect all radical processes—intramolecular and others occurring in bulk, including diffusion quenching of Trp in an excited state by Trp in the ground state. Polarized signals of the following protons have been observed for peptides **I**–**III**: aromatic 12, 8 and methylene 7 of the initial compounds. In addition, methyl protons of the N-acetyl group (NHCOCH_3_) in **I** and **II** and proton signals of products (8.3–8.4 ppm) formed from peptides **II** and **III** are polarized (Appendix A).

The CIDNP effects analysis of aromatic and methylene protons of Trp moiety in peptides, according to the existing rules for high magnetic fields, corresponds to the occurrence of CIDNP in the triplet pair of the Trp radical-cation and the radical-anion with a paramagnetic center located at the carbonyl group of the peptide bond. This refers to the bond assigned on the N-end of the peptide (Figure 1). Such a radical pair can be formed due to diffusion quenching of the Trp triplet excited state by another Trp moiety in the ground state. The conclusion about the multiplicity of the reactive state of the peptide was made based on the analysis of the CIDNP signs [20,21]. Details of the analysis are presented in Appendix A.

Thus, the analysis shows that the observed CIDNP effects are not from BZ, formed due to intramolecular ET. The concentration dependencies of the CIDNP coefficients for all polarized protons confirm this conclusion (Figure 5 and Appendix A). To obtain the CIDNP coefficients shown in the figures, we had to calculate which fraction of the signal integral intensity in the NMR spectrum after photolysis corresponds to the initial peptide involved in the process of diffusion quenching of the Trp excited state by the carbonyl group of the peptide in the ground state (Appendix A).

Concentration dependences of the CIDNP effects for peptides **I**–**II** shown in Figure 5 confirm the CIDNP origin suggestion. As a result of the approximation of CIDNP dependence on concentration as a reaction by a pseudo-first-order, we obtained the ratio of ET rate constants in I and II equal 1.65. This conclusion about the L- and D-isomers’ activity difference also correlates with the fluorescence quenching results (Figure 2). In this case, D-isomer **II** is quenched more slowly than both L-analogs: **I** and acetate salt **III**. Since fluorescence quenching is likely to occur through the ET mechanism similar to diffusion quenching of the triplet excited state, it can be concluded that the D optical isomer is less active in the ET process.

Regarding acetate salt **III**, the rapid plateauing of the curve in Figure 5b is likely to result from the counter-ion influence [22]. Thus, the above results produced experimental evidence of ET between the peptide bond and the Trp moiety in short peptides [2,8,23,24,25]. Note that the reference data indicates intramolecular ET, whereas the TR CIDNP in a high magnetic field in peptides **I**–**III** is absent due to the probable presence of an electron exchange interaction in BZ caused by the short distance between donor and acceptor moieties [13].

Nevertheless, the observation of CIDNP suggests that the peptide bond has acceptor properties attributed to it in the literature [2,6,7,23,24,25]. The conclusion about the possibility of ET between amino acids and the CONH bond is mainly based on the analysis of tryptophan and histidine dependencies of fluorescence quantum yields on the microenvironment [2,23,24,25]. This concerns the properties of the nearest groups in a protein or peptide, including charged ones, the length of the chain between the donor and acceptor, and the solvent polarity and pH. There are even examples of a quantitative correlation between the values of the experimental Trp fluorescence quantum yields and those calculated within the framework of the ET model [24]. As a radical pair, in this case, was considered the highest occupied molecular orbital (MO) localized on the Trp indole ring and the lowest free p* MO of the amide group under conditions of strong Coulomb interaction. Thus, the evidence of ET in peptides available so far was indirect while the CIDNP data directly indicate the formation of the Trp radical-cation and the CONH radical-anion.

In addition, we have mentioned above an intense negative polarization of the methyl protons of the N-acetyl group (NHCOCH_3_) in peptides **I** and **II**. The concentration dependence of the CIDNP coefficients of methyl protons for the L- and D-configurations of the peptide is shown in Figure 6.

Since the tangent of this dependence (0.007) differs noticeably from those for protons 12 and 7 in Figure 5 (0.038; 0.023; 0.013), it is evident that these dependencies originate from different processes. The CIDNP of methyl protons is assumed to be formed in the radical pair with N-acetyl moiety: NHCOCH_3_ and Trp residue (Figure 6). It is worth noting that, in this case, chiral centers do not influence the parameters of Trp singlet excited state quenching. The fact that the singlet excited state is quenched follows from the analysis of the CIDNP sign (S) of the methyl protons of the N-acetyl group (details see Appendix A). The possibility of ET between these peptide moieties was considered in work [22] when calculating the efficiency of ET in N-acetyltryptophanamide. The estimated spin density distribution in the NHCOCH_3_ radical-anion is presented in Appendix A.

The short lifetime of the Trp singlet excited state allows us to assume that, in this case, ET occurs under conditions of the formation of a weak collisional complex. It also might be the hydrogen bond between two CONH groups described in the literature [14,15].

## 3. Materials and Methods

### 3.1. Spectroscopic Measurements

All UV spectroscopic measurements were performed using a quartz cuvette of 1 cm optical length. Spectra and kinetic luminescence curves were recorded with an Edinburgh Instruments FLSP-920 spectrofluorimeter (Livingston, UK) with either a Xenon lamp or laser diodes EPLED-270 (λ_ex_ = 270 nm, pulse duration 0.6 ns) as excitation sources. The kinetic traces were fitted with exponential decay functions using a reconvolution procedure (FAST program–version 3.5.2). The fluorescence quantum yields of peptides were determined relative to Trp as standard fluorophore [26] using the following equation [27]:$$\varphi_{u}=\frac{A_{s}F_{u}n^{2}}{A_{u}F_{s}n_{0}^{2}}\times \varphi_{s}$$
where, *φ_s_*—quantum yield of standard fluorophore (0.14 [26]), *φ_u_*—quantum yield of unknown fluorophore, *A_s_*—the absorbance of standard fluorophore at the excitation wavelength, *A_u_*—the absorbance of unknown fluorophore at the excitation wavelength, *Fs*—the area of integrated fluorescence intensity of the reference sample when excited at the same excitation wavelength, and *F_u_*—the total area of integrated fluorescence intensity for the unknown sample when excited at the same excitation wavelength. The refractive indices of the solvents for the unknown and the standard samples are denoted by n_u_ and n_s_, respectively. All fluorescence measurements of isoabsorptive solutions of the standard and peptides were performed in the same solvent–water mixture. The absorption spectra were recorded using an Agilent 8453 spectrophotometer (Santa Clara, CA, USA). All experiments were performed at room temperature 296 K.

### 3.2. NMR and Photo-CIDNP Measurements

Lys-L-Trp-Lys acetate salt (Bachem, Bubendorf, Switzerland, 99.3%), Lys-L/D-Trp-Lys with N-terminal acetylation (Elabscience, Houston, TX, USA, 95.3%) and the deutero solvent water-d_2_ (99.9% D, Cambridge Isotope Laboratories, Andover, MA, USA) were used as received. A Bruker Avance HD III NMR spectrometer (500 MHz ^1^H operating frequency, τ(π/2) = 10 μs) was used to record the ^1^H NMR spectra of the photoproducts. A Bruker DPX200 NMR spectrometer (200 MHz ^1^H operating frequency, τ(π/2) = 2.5 μs) was utilized to study the CIDNP effects. The Lambda Physik EMG 101 MSC excimer laser (Göttingen, Germany) was used as a light source (308 nm, 100 mJ, pulse duration 15 ns) in these experiments. The samples were bubbled with argon for 15 min to remove dissolved oxygen and were irradiated directly in the probe of the DPX200 NMR spectrometer. Time-resolved (TR) CIDNP experiments were performed with the following pulse sequence: presaturation, laser pulse (−15 ns), variable time delay, and π/2 radio-frequency registration pulse. The pseudo-steady state (PSS) experiments were proposed to increase the signal-to-noise ratio [19]. In PSS experiments, a standard pulse sequence was used: presaturation, delay 1, pulse τ(π), delay2 (16 laser flashes with repetition rate 50 Hz during delay 2), observation pulse τ(π/2), and acquisition. Delay1/delay2 ≈ 1.1 to remove residual signals of solvents and solutes. After laser irradiation, the ^1^H NMR spectra of products were recorded. All experiments were performed at room temperature, 296 K.

## 4. Conclusions

Thus, a joint analysis of peptides **I**–**III** fluorescence quenching data and the CIDNP effects formed upon UV irradiation allowed us to arrive at the following conclusions. The data on fluorescence quenching and the relative rates of bimolecular quenching of the triplet excited state of Trp in peptides **I**–**III** indicate a greater efficiency of ET in the L-optical isomer than in the D-analog. The manifestation of CIDNP effects in bimolecular processes implies that quenching occurs via the ET mechanism. ET was shown to occur between the donor—Trp moiety while the CONH peptide bond acts as an acceptor. It should be noted that CIDNP detecting on Trp protons is the first, although not direct, confirmation of the paramagnetic form of the CONH group. Another confirmation of the electron-withdrawing properties of the N-acetyl group NHCOCH_3_ is the presence of CIDNP on its methyl protons in peptides **I** and **II** under UV irradiation.

Moreover, considerable differences are observed in the photoinduced transformations of peptides **I**–**III**. ^1^H NMR analysis of the products shows that L-optical isomer **I** undergo less photodecomposition (15% ± 10%) than salt **III** and D-analog **II**. It seems to be a consequence of the high reversibility of ET through which, according to the proposed schemes, the photocleavage products are formed, whereas in the case of salt **III** with the same optical configuration, the formation of a cyclic product is assumed. In addition, the products of CONH and C^6^-C^7^ bonds cleavage—aldehyde and 3-methyleneindolinine—were observed. The photolysis of the D-optical isomer **II** also results in the formation of aldehyde and 3-methyleneindolinine. Various photocleavage products formed from **I** and **II** point to the difference in the reactivity of the L- and D-optical isomers. It is worth mentioning that the same conclusion was drawn from the analysis of spin effects in chiral dyads with L- and D-Trp isomers. Spin effects reflect differences in spin density distribution in paramagnetic precursors, which correlate with the reactivity of compounds in radical processes. In addition, the variety of products shows that the reactivity of these peptides depends not only on the optical configuration, but also on a counter-ion presence. Thus, replacing the amide group in the first position of peptide **I** with a donor amino group in its salt **III** results in a radical cleavage of the CONH and C^6^-C^7^ bonds.

## Data Availability

Not applicable.

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
