# Peer review of "Photoinduced Processes in Lysine-Tryptophan-Lysine Tripeptide with L and D Tryptophan"

_ijms, 2023, doi:10.3390/ijms24043331_

Round 1

Reviewer 1 Report

A major revision is Needed.

Why acetate salts? There is various kind of slats available, including TFA, and HCl salt. The authors should clarify. The authors can have a look at the following paper with a proper citation: Sci Rep 6, 32670 (2016)

Why NHCO, not CONH?

The authors should add the full form of the amino acid when they are introducing 3-letter amino acid for the first time.

Scheme 2: Figure quality is bad. The authors are requested to improve the quality of the figure and presentation.

The authors should add the proper chemical structure of the peptide. The lysine side chain in Isomer III is out of the page. It is true for Figure 4 also. Please correct it.

The authors are requested to add the quantum yield. The authors did not mention how did they calculate the quantum yield.  The authors can have a look at the following paper with a proper citation for quantum yield calculation: J. Phys. Chem. C 2018, 122, 6, 3655–3661

The authors are requested to add the table containing all decay parameters related to decay measurements. Phys. Chem. Chem. Phys., 2003,5, 1386-1391

H NMR investigation of Lys-Trp-Lys photolysis products: I think the author should write 1H NMR.

The authors must format the reference section according to the journal criteria. For example, Front Bioeng Biotechnol is to be changed to Front. Bioeng. Biotechnol.; Chemistry – A European Journal to be changed to Chem. Eur. J.

Reviewer 2 Report

In this paper, Ageeva et al. investigated the photoinduced intramolecular and intermolecular processes of Ac-Lys-L/D-Trp-Lys and Lys-L-Trp-Lys. Interestingly, the authors identified the effect of the L and D conformations of Trp on the fluorescence quantum yield of the peptide, and the resulting photocleavage products. Overall, this work has performed well in detailed experimental studies and mechanistic analysis and has important implications for the study of photo-sensitive peptide supramolecular materials and photobiology. I would like to recommend it for publication in IJMS after the following points can be addressed.

1. Why did the authors choose Ac-Lys-L/D-Trp-Lys and Lys-L-Trp-Lys instead of introducing Lys-D-Trp-Lys into the study?

2. The authors used acetate counter ion in this study, does the different anion affect the photoinduced process results?

3. In this manuscript, some figures have been cropped and cannot clearly presented, and the author should be aware of this when making changes.

4. what is the temperature of the photolysis experiment, how can temperature interference with peptide samples be excluded?

5. Reference selection is good in the manuscript. It is recommended to add recent articles on design strategies and applications of short peptide-based supramolecular materials. (e.g., 10.1021/jacs.1c11750; 10.1002/anie.202105830).

Round 2

Reviewer 1 Report

The manuscript can be accepted after adding a detailed procedure for quantum yield measurement with proper citation. The authors are requested to repeat the decay experiments as the ki square value is more than one. It is very difficult to accept the data.

Author Response

As requested, we have added a detailed procedure for quantum yield measurement and repeated the decay experiments. Changes are marked by color.

Finally, the authors thank you for your comments, which allowed us to improve the article.